# NoRA: Nested Low-Rank Adaptation for Efficient Fine-Tuning Large Models

## Abstract

Low-Rank Adaptation (LoRA) has become a popular paradigm for fine-tuning large models, but it still necessitates a substantial number of training parameters. To address this issue, we first conduct comprehensive empirical studies on parameter-efficient LoRA structure. Then, we establish design guidelines that emphasize the use of serial structures, optimal placements, and nested LoRA. Based on these insights, we present NoRA, a nested parameter-efficient LoRA structure that revolutionizes the initialization and fine-tuning of projection matrices. Our NoRA's innovative approach involves freezing outer layer LoRA weights and employing a serial inner layer design, enabling precise task-specific adaptations while maintaining compact training parameters. In addition, we propose an activation-aware Singular Value Decomposition (AwSVD) that adjusts the weight matrices based on activation distributions for initialization of outer layer LoRA weights. This schema enhances decomposition accuracy and mitigates computational errors. Extensive evaluations across multiple linguistic and visual tasks demonstrate that NoRA outperforms state-of-the-art LoRA variants, achieving significant improvements in efficiency and effectiveness on models such as Mistral-7B, Gemma-7B, and LLaMA-3 8B. Notably, NoRA reduces fine-tuning parameters|training-time|memory-usage by 85.5%|37.5%|8.9% and enhances performance by 1.9%, compared to LoRA on LLaMA-3 8B. Codes are available in the supplementary materials.

## 1 Introduction

Large Language Models (LLMs) have recently achieved remarkable performance in natural language processing and related fields (Zhao et al., 2023; Touvron et al., 2023). However, the high parameter size makes training and adaptation challenging, especially in resource-limited settings. To address this, Parameter-Efficient Fine-Tuning (PEFT) techniques have been developed (Ding et al., 2023; Han et al., 2024), focusing on fine-tuning a subset of model parameters. Low-Rank Adaptation (LoRA) (Hu et al., 2021a) is a notable PEFT technique that uses low-rank matrices for efficient adaptation to specific tasks (He et al., 2021). It achieves significant computational and memory savings during fine-tuning, making it feasible to adapt LLMs on consumer-grade hardware (Mao et al., 2024).

Despite LoRA's demonstrated utility, it faces challenges that limit its effectiveness in downstream tasks. The original LoRA involves training a large number of parameters, which can lead to slow convergence and potential overfitting problems. To address these issues, two main approaches have emerged in the literature: (1) Hyperparameter-based methods, which focus on adaptive rank allocation and optimization settings tuning. Examples include BiLoRA (Qiang et al., 2024), LoRA-dropout (Lin et al., 2024), and AdaLoRA (Zhang et al., 2023b), which employ bi-level optimization strategies, parameter dropout, and singular value-based allocation for different layer types. (2) Structural modifications, which involve new components or frozen architectures. For instance, DoRA (Liu et al., 2024b) and SARA (Gu et al., 2024) augment vector and mixture designs, respectively, although at the expense of increased computational costs. VeRA (Liu et al., 2023a) incorporates trainable vectors on random matrices, while other methods (Bałazy et al., 2024) approximate SVD decomposition and selectively truncate singular values to balance performance and efficiency. Despite these advancements, two significant challenges persist for these LoRA variants: (1) The intrinsic properties of LLMs are often neglected, particularly their sensitivity to activation outliers, which can potentially lead to

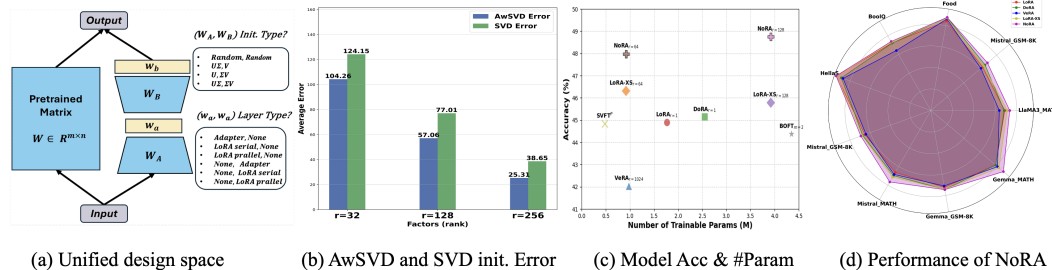

| (a) Unified design space | (b) AwSVD and SVD init. Error | (c) Model Acc & #Param | (d) Performance of NoRA |
| --- | --- | --- | --- |

Figure 1: Figure (a) illustrates the configurations of different architectural modifications explored in this study, highlighting the design locations and initialization strategies for layer adaptations. Figure (b) compares the errors of SVD and AwSVD, while Figures (c) and (d) compare other baseline methods.

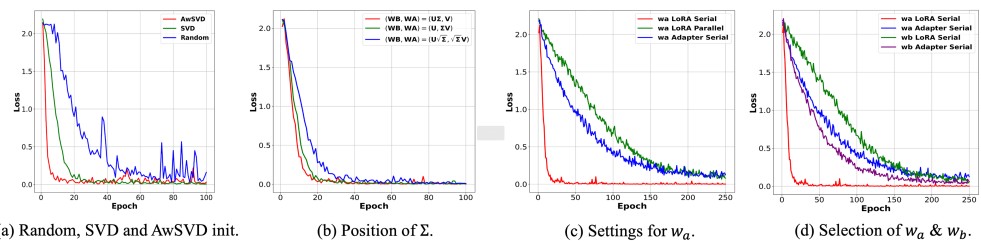

| (a) Random, SVD and AwSVD init. | (b) Position of $\Sigma$. | (c) Settings for $w_a$. | (d) Selection of $w_a$ & $w_b$. |
| --- | --- | --- | --- |

Figure 2: Figures (a), (b), (c), and (d) depict the loss curves for various architectural configurations of the CLIP model on the DTD dataset during training, highlighting the specific impacts of different initialization methods (random, SVD, AwSVD) and layer adaptation strategies (Adapter, LoRA serial and parallel, placement strategies).

substantial decomposition errors. (2) There is a lack of a unified design and evaluation framework for initialization strategies and trainable structures.

To address these challenges, we conduct a comprehensive analysis of recent variants such as VeRA and LoRA-XS (Zhang et al., 2023a), observing that they fundamentally design trainable structures (*e.g.*, adapters) for frozen low-rank matrices. Building on this insight, we investigate LoRA as a trainable structure in both parallel and serial forms. As illustrated in Figure 1 (a), our construct unified design space encompasses various initialization strategies and trainable structure options. Through empirical exploration of this design space, we derive several key insights: (1) Regarding initialization, we find that SVD consistently outperforms random initialization. Furthermore, we introduce an activation-aware Singular Value Decomposition (AwSVD) technique to further accelerate convergence (see Figure 2 (a)). (2) We investigate scenarios where singular vectors are contained in different matrices of the SVD ($W_A$, $W_B$, $W_A$ & $W_B$ in Figure 2 (b)) with varying trainable structures and positions ($w_a$ or $w_b$ in Figure 2 (c)). Our findings reveal that although the three singular vector locations exhibit similar performance, faster convergence is achieved when they are contained in $W_A$. (3) As shown in Figure 2 (d), LoRA serial stably demonstrates superior performance compared to adapter serial and LoRA parallel across three distinct scenarios. Additionally, we observe that $w_a$ proves to be a more advantageous position than $w_b$ for augmenting trainable parameters. These empirical observations provide a foundation for the development of more effective and efficient LoRA variants that address current limitations and leverage the unique properties of LLMs. In brief, we explore various architectural modifications, including parallel and serial adapters, nested LoRA, and design placements ($(W_A, W_B)|(w_a, w_b)$) to enhance fine-tuning strategies. Through extensive empirical research, we derive valuable design guidelines for optimizing the configuration of LoRA. Specifically, we propose the following guidelines: 1) SVD initialization plays a crucial role in enhancing the effectiveness of LoRA structure design; 2) In the unified design space, $w_a$ should be favored over $w_b$ for superior performance; 3) It is recommended to configure LoRA as a serial structure rather than a parallel one, and to prefer nested LoRA over traditional adapters for improved fine-tuning efficiency.

Based on the above guidelines, we propose NoRA, a nested parameter-efficient LoRA design structure. It features a nested LoRA structure in which the outer LoRA is initialized using AwSVD, while the serial inner LoRA layers are initialized with a Gaussian distribution. NoRA aims to enhance the efficiency and effectiveness of LoRA by optimizing the initialization of projection matrices and fine-tuning strategies. NoRA keeps the outer LoRA fixed while innovatively reducing the number of parameters and maximizing adaptation performance. **First**, NoRA introduces a new initialization method for the LoRA projection matrices. We propose AwSVD to decompose the original matrices, effectively reducing output errors while maintaining high fidelity to the pre-trained weights. This initialization strategy provides a more informed starting point for the fine-tuning process, helping to accelerate convergence and improve task-specific performance (see Figure 2 (a) and Figure 1 (c), (d)). **Second**, NoRA effectively reduces training parameters by freezing the outer LoRA weights while employing a serial inner LoRA design, enabling the model to adapt more precisely to specific tasks while maintaining a compact parameter space.

We conduct experiments on multiple downstream tasks, including instruction tasks on the GSM8K (Cobbe et al., 2021) and Math (Hendrycks et al., 2021) datasets using the Mistral-7B (Jiang et al., 2023), Gemma-7B (Team et al., 2024), and LLaMA-3 8B models. Additionally, we fine-tune the LLaMA (Touvron et al., 2023) model for commonsense reasoning, perform few-shot tuning on the CLIP (Radford et al., 2021) model, and conduct subject-driven generation on the Stable Diffusion XL (Podell et al., 2023) model. In these experiments, NoRA not only significantly reduces the required parameters to as low as 4.1 million for the LLaMA-3 8B model but also enhances performance, achieving an average score of 84.4%, which surpasses LoRA's 82.8%. Furthermore, in visual few-shot tasks using ViT-B/16, NoRA achieves the highest average accuracies of 80.9% (4 shots) and 86.1% (16 shots), demonstrating its superior efficiency and effectiveness over existing methods. We summarize our contributions as follows:

- To overcome limitations of existing methods, we construct a unified design space while maintaining a compact parameter set. Through comprehensive empirical research, we develop a set of design guidelines that emphasize the importance of design positions ($W_A|w_a$), serial structures, and the use of nested LoRA.

- We propose an AwSVD technique that adjusts weight matrices based on activation distributions, effectively managing activation outliers and accelerating model convergence.

- We introduce NoRA, the first nested LoRA structure that optimizes the initialization and fine-tuning of projection matrices. NoRA offers key advantages: significant parameter reduction, enhanced training efficiency, and improved performance across diverse tasks.

- Through extensive evaluations across various linguistic and visual tasks, we demonstrate NoRA's superior performance, highlighting improvements in efficiency and effectiveness compared to state-of-the-art LoRA variants.

## 2 RELATED WORK

Parameter-efficient fine-tuning (PEFT) (Pfeiffer et al., 2020; Zaken et al., 2021) emerges as an effective solution for adapting large pre-trained models to downstream tasks, successfully addressing the challenges of high computational demands and training costs associated with traditional fine-tuning methods (Hu et al., 2023). PEFT optimizes parameter adjustment by reducing additional parameters and computational resources for specific tasks while maintaining the structure and performance of the pre-trained model. The field evolves from early selective update strategies (Gururangan et al., 2020) to more advanced techniques such as adapter modules and delta-weight methods. These innovative approaches include adapters (Houlsby et al., 2019), which introduce task-specific parameters within transformer layers. Additionally, prompt tuning (Liu et al., 2023b) and prefix tuning (Li & Liang, 2021) adapt to tasks by appending task-specific vectors to inputs or various layer representations. BitFit and IA3 (Zaken et al., 2021; Liu et al., 2022) focus on altering only the bias or scaling vectors within the base large language model. Overall, these methods, including LoRA (Hu et al., 2021a) and OFT (Liu et al., 2023c), aim to further enhance model adaptability through streamlined updates and auxiliary modules.

Low-rank Adaptation (LoRA) proves efficient in various task scenarios, using low-rank decomposition to enhance adaptation while minimizing computational overhead. However, its fixed rank

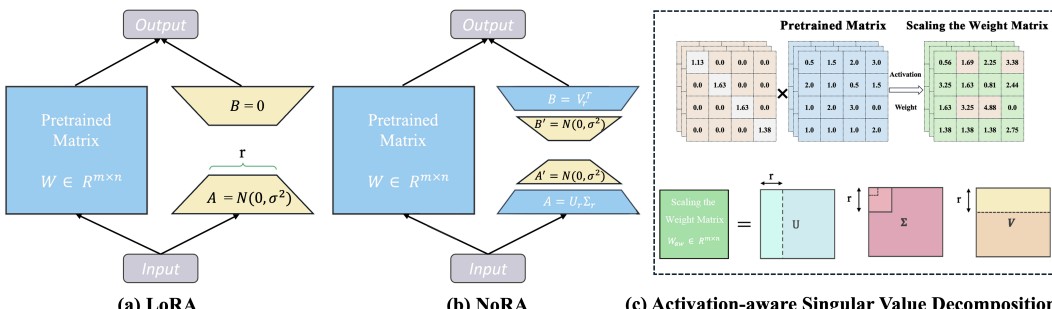

(a) LoRA      (b) NoRA      (c) Activation-aware Singular Value Decomposition

Figure 3: (a) LoRA structure; (b) In the NoRA structure, the outer LoRA ($A|B$) is initialized using AwSVD, while the inner LoRA ($A'|B'$) is initialized with a Gaussian distribution. The blue modules represent the frozen weights, while the yellow modules indicate the components that require updates. (c) Details of the AwSVD process. Here, $r$ denotes the outer rank, and "Scaling the weight matrix" refers to the matrix awaiting decomposition after weight activation.

limits flexibility in diverse tasks. Researchers propose LoRA variants to address these limitations. AdaLoRA (Zhang et al., 2023b) employs singular value decomposition to parameterize incremental updates to the pretrained weight matrices, striking a balance between adaptation fidelity and the preservation of pre-existing knowledge structures. LoRA-FA (Zhang et al., 2023a) reduces activation memory by freezing partial weights but remains rank-limited. VeRA (Liu et al., 2023a) enhances scalability but remains sensitive to hidden dimensions. LoRA-XS (Zhang et al., 2023a) improves real-time performance and memory efficiency but does not fully address task-specific complexity. PiSSA (Meng et al., 2024) selectively adjusts matrix ranks and distributions, enhancing large-scale model applicability in complex tasks. Additionally, DoRA (Liu et al., 2024b) optimizes LoRA by improving parameter efficiency and the matrix update structure. FLoRA (Si et al., 2024) introduces an extra core based on Tucker decomposition to maintain a consistent topological structure. MoSLoRA (Wu et al., 2024) incorporates a learnable mixer to flexibly fuse subspace information. Although all three methods enhance adaptability, they also lead to increased training costs. Compared to the aforementioned improvements, our main advantage lies in designing a unified search space to find a simple yet effective method. By introducing NoRA, we aim to optimize the initialization and fine-tuning strategies of the LoRA projection matrix. Additionally, we propose an AwSVD method that effectively reduces output errors and decreases the number of training parameters by freezing the outer LoRA weights.

## 3 METHODOLOGY: NESTED LOW-RANK ADAPTATION

### 3.1 REVIEW OF LOW-RANK ADAPTATION

LoRA is a parameter-efficient method for fine-tuning large-scale pre-trained models. It achieves fine-tuning of the original weights $\mathbf{W}$ by introducing low-rank matrix updates, aiming to preserve the stability and overall performance of the pre-trained models. The traditional LoRA forward pass for an input $x \in \mathbb{R}^n$ is:

$$h = \mathbf{W}x + \Delta\mathbf{W}x = \mathbf{W}x + \mathbf{B}\mathbf{A}x, \tag{1}$$

where $\Delta\mathbf{W} \in \mathbb{R}^{m \times n}$ is the low-rank weight update, and $\mathbf{A} \in \mathbb{R}^{r \times n}$ and $\mathbf{B} \in \mathbb{R}^{m \times r}$ are low-rank matrices with $r \ll \min(m, n)$. During training, we keep $\mathbf{W}$ frozen, while $\mathbf{A}$ and $\mathbf{B}$ serve as the trainable parameters.

### 3.2 NORA STRUCTURE AND INITIALIZATION

As illustrated in Figure 3, NoRA initializes using activation-aware singular value decomposition (AwSVD) and employs a nested Low-Rank Adaptation (LoRA) architecture.

The forward path of NoRA for an input $x \in \mathbb{R}^n$ is expressed as:

$$h = \mathbf{W}x + \Delta\mathbf{W}x = \mathbf{W}x + \mathbf{B}\mathbf{B}'\mathbf{A}'\mathbf{A}x, \tag{2}$$

where $A$ and $B$ represent the outer LoRA matrices, and $A'$ and $B'$ denote the inner LoRA matrices. The specific details are as follows:

- **Outer LoRA Layer**: The LoRA weights for this layer are initialized using the activation-aware SVD of the pre-trained weights $\mathbf{W}$, with the decomposition error mitigated by a scaling matrix $\mathbf{S}$. Specifically, matrix $\mathbf{B}$ is initialized with $\mathbf{U}\mathbf{\Sigma}$, while matrix $\mathbf{A}$ is initialized with $\mathbf{V}^{\mathbf{T}}\mathbf{S}^{-1}$. The parameters of this outer LoRA layer are frozen during training to maintain stability and preserve the essential features of the pre-trained model, while still permitting precise adjustments through the inner LoRA layer.

- **Inner LoRA Layer**: This layer is initialized with a Gaussian distribution $N(0, \sigma^2)$. Such initialization enables the inner LoRA layer to focus on subtle perturbations within the weight space, facilitating finer adjustments without altering the core weights preserved by the outer LoRA layer. This approach ensures that updates are concentrated on refining and enhancing the model's ability to adapt to new tasks, leveraging minor adjustments that have a targeted impact on performance.

### 3.3 ACTIVATION-AWARE SINGULAR VALUE DECOMPOSITION

To enhance the effectiveness of LoRA initialization, we incorporate activation information into the Singular Value Decomposition (SVD) process. This strategy arises from the observation that not all weights contribute equally to the model's output; their significance can be more accurately estimated by considering their interaction with typical input activations. Let $\mathbf{X} \in \mathbb{R}^{b \times n}$ represent a batch of input activations, where $b$ denotes the batch size. The activation-weighted matrix is defined as follows:

$$\mathbf{S} = \text{diag}\left(\sqrt{\frac{1}{n}\sum_{j=1}^{n}|\mathbf{X}_{:,j}|}\right), \tag{3}$$

**Algorithm 1** PyTorch code for NoRA

```python
# r_out: rank of the outer LoRA layer.
# BB'A'A represents the weight of NoRA.

def init_nora_param(W, r_out):
    S_d = torch.diag(torch.mean(
                        torch.abs(W)))
    U, S, V = torch.svd(W @ S_d)
    B = U[:, :r_out] @ torch.diag(S[:r_out])
    A = V.T[:r_out, :] @ torch.inverse(S_d)

def forward(x):
    output = F.linear(x, W) + BB'A'Ax
```

$$\mathbf{W_{aw}} = \mathbf{W_{original}} \cdot \mathbf{S}, \tag{4}$$

where $\mathbf{W_{aw}} \in \mathbb{R}^{m \times n}$ represents the activation-weighted matrix, and $\text{diag}(\cdot)$ denotes a function that creates a diagonal matrix from a vector. This weighting scheme aligns the adaptation process with the characteristics of the input data, emphasizing the most impactful parameters and potentially enhancing the overall effectiveness of the fine-tuning process. Building on the activation-weighted SVD, we define the NoRA initialization method, which integrates the advantages of SVD-based initialization with activation-guided weighting. The specific formulation is as follows:

$$\mathbf{W_{aw}} = \mathbf{U}\mathbf{\Sigma}\mathbf{V}^T, \tag{5}$$

$$\mathbf{B} = \mathbf{U}[:, :r]\mathbf{\Sigma}[:r, :r], \quad \mathbf{A} = \mathbf{V}^{\mathbf{T}}[:r, :]. \tag{6}$$

We obtain the sensitivity of the weights to the input through an activation-aware matrix, and we use SVD to maximally preserve this information in the frozen outer LoRA weights $\mathbf{A}$ and $\mathbf{B}$. Additionally, to reduce the error in the activation output compared to the original activation weight $\mathbf{W}$, we multiply $\mathbf{A}$ by the inverse of the scaling matrix $\mathbf{S}$:

$$\mathbf{W_{original}} \approx \mathbf{B}(\mathbf{A}\mathbf{S}^{-1}) = \mathbf{U}[:, :r]\mathbf{\Sigma}[:r, :r](\mathbf{V}^{\mathbf{T}}[:r, :]\mathbf{S}^{-1}), \tag{7}$$

where $\mathbf{S} \in \mathbb{R}^{n \times n}$ is the diagonal matrix of activation standard deviations.

## 3.4 Understandings of NoRA Structure

For better understanding, we provide comparisons between our NoRA and alternative approaches such as adding adapters or parallel LoRA structures:

$$\text{Adapter:} \quad h = \mathbf{W}x + \mathbf{BRA}x, \quad \text{Parallel LoRA:} \quad h = \mathbf{W}x + (\mathbf{B} + \mathbf{CA}')\mathbf{A}x, \qquad (8)$$

where $\mathbf{W} \in \mathbb{R}^{m \times n}$, $\mathbf{A} \in \mathbb{R}^{r \times n}$, $\mathbf{B} \in \mathbb{R}^{m \times r}$, $\mathbf{A}' \in \mathbb{R}^{r' \times r}$, $\mathbf{B}' \in \mathbb{R}^{r \times r'}$, $\mathbf{C} \in \mathbb{R}^{m \times r'}$ and $\mathbf{R} \in \mathbb{R}^{r \times r}$. The NoRA form provides a more expressive and flexible weight update compared to adding adapter or parallel LoRA structures. We analyze the expressiveness, flexibility, and parameter efficiency of each form:

**Expressiveness**: The weight updates for each form can be expressed as:

$$\mathbf{\Delta W_{NoRA}} = \mathbf{BB}'\mathbf{A}'\mathbf{A}, \quad \mathbf{\Delta W_{Adapter}} = \mathbf{BRA}, \quad \mathbf{\Delta W_{Parallel}} = (\mathbf{B} + \mathbf{CA}')\mathbf{A}. \qquad (9)$$

NoRA introduces a nested low-rank structure that allows for more complex transformations of the input space. To show this, we can consider the rank of each update:

$$\text{rank}(\mathbf{\Delta W_{NoRA}}) \leq \min(r, r'), \quad \text{rank}(\mathbf{\Delta W_{Adapter}}) \leq r, \quad \text{rank}(\mathbf{\Delta W_{Parallel}}) \leq r. \qquad (10)$$

While the rank of NoRA is bounded by $\min(r, r')$, its nested structure allows for more complex non-linear transformations within this rank constraint.

**Parameter Efficiency**: The number of additional parameters for each form is:

$$\mathbf{P_{NoRA}} = rr' + r'r, \quad \mathbf{P_{Adapter}} = r^2, \quad \mathbf{P_{Parallel}} = mr' + r'n. \qquad (11)$$

NoRA introduces a controlled number of additional parameters through its nested structure, allowing for a flexible trade-off between expressiveness and efficiency by adjusting $r$ and $r'$.

**Flexibility**: NoRA's nested structure $(\mathbf{BB}')(\mathbf{A}'\mathbf{A})$ allows for separate optimization of the outer ($\mathbf{B}$ and $\mathbf{A}$) and inner ($\mathbf{B}'$ and $\mathbf{A}'$) layers. This separation enables the model to learn both coarse and fine-grained adaptations simultaneously. In contrast, the adding adapter form $\mathbf{BRA}$ and parallel LoRA form $(\mathbf{B} + \mathbf{CA}')\mathbf{A}$ lack this hierarchical structure, limiting their ability to capture multi-scale adaptations.

**Generalization**: NoRA can be seen as a generalization of both the adding adapter and parallel LoRA forms:

- By setting $\mathbf{B}' = \mathbf{R}$ and $\mathbf{A}' = \mathbf{I}$, where $\mathbf{I}$ is the identity matrix, NoRA reduces to the adapter form.
- By setting $\mathbf{B}' = \mathbf{I}$ and rearranging terms, NoRA can approximate the parallel LoRA form.

The generalization capability of NoRA enables flexible adaptation to diverse scenarios, potentially harnessing the strengths of both approaches. The NoRA architecture integrates the expressiveness of both the additive adapter and parallel LoRA configurations while providing additional flexibility and facilitating multi-scale adaptations. As previously analyzed, the nested structure of NoRA is inherently flexible, allowing it to manage complex multi-scale adaptations within a controlled parameter space. Moreover, NoRA's generalization capability permits structural simplification when necessary, enabling adaptation to various fine-tuning scenarios and enhancing its versatility.

## 4 Experiment

In this section, we provide detailed descriptions of our experiments evaluating the effectiveness of the NoRA method. We begin with instruction tuning experiments on the Mistral-7B, Gemma-7B, and LlaMA-3 8B models to evaluate NoRA's capability to enable large language models (LLMs) to follow instructions with minimal parameter overhead. Next, we examine the reasoning capabilities of NoRA in comparison to other benchmark methods (Hu et al., 2021b; Liu et al., 2024a; 2023a;

Bałazy et al., 2024) on common-sense reasoning tasks using the Llama-3 8B model. Furthermore, we investigate the generalization and adaptability of NoRA in the domains of vision-language models and theme-driven generation. Finally, we analyze two SVD decomposition techniques and structural design guidelines, providing a detailed comparison of NoRA's training time, GPU memory usage, and loss curves relative to LoRA and other benchmark methods.

Table 1: Instruction Tuning Performance on GSM8K and MATH Benchmarks for Mistral-7B, Gemma-7B, and Llama-3 8B Models using Full Fine-tuning, LoRA, DoRA, VeRA, LoRA-XS, and NoRA.

| Method | Mistral-7B | | | | Gemma-7B | | | | LLaMA-3 8B | | | |
|---|---|---|---|---|---|---|---|---|---|---|---|---|
| | #Params | GSM-8K | MATH | AVG | #Params | GSM-8K | MATH | AVG | #Params | GSM-8K | MATH | AVG |
| Full-FT | 7.2B | 67.02 | 18.60 | 42.81 | 8.5B | 71.34 | 22.74 | 47.04 | 8.0B | 64.13 | 16.24 | 40.19 |
| LoRA$_{r=64}$ | 168M | 67.70 | 19.68 | 43.69 | 200M | 74.90 | 31.28 | 53.09 | 168M | 76.25 | 24.92 | 50.89 |
| LoRA$_{r=1}$ | 1.77M | 65.38 | 16.57 | 40.98 | 0.82M | 72.40 | 26.28 | 49.34 | 1.77M | 68.84 | 20.94 | 44.89 |
| DoRA$_{r=1}$ | 2.55M | 67.54 | 17.43 | 42.49 | 3.26M | 74.37 | 26.28 | 50.33 | 2.55M | 68.30 | 21.96 | 45.13 |
| VeRA$_{r=1024}$ | 0.98M | 64.32 | 17.13 | 40.73 | 0.43M | 71.11 | 27.04 | 49.08 | 0.98M | 63.76 | 20.28 | 42.02 |
| LoRA-XS$_{r=64}$ | 0.92M | 68.01 | 17.86 | 42.94 | 0.80M | 74.22 | 27.62 | 50.92 | 0.92M | 71.19 | 21.43 | 46.31 |
| LoRA-XS$_{r=128}$ | 3.92M | 67.83 | 18.12 | 42.97 | 3.21M | 71.56 | 25.24 | 48.40 | 3.92M | 71.27 | 20.24 | 45.78 |
| NoRA$_{r=64}$ | 0.92M | 69.39 | 19.14 | 44.27 | 0.80M | 74.60 | **29.40** | 51.93 | 0.92M | 73.46 | 22.94 | 48.20 |
| NoRA$_{r=128}$ | 3.92M | **70.92** | **19.83** | **45.38** | 3.21M | **74.90** | 29.22 | **52.06** | 3.92M | **73.62** | **23.88** | **48.75** |

## 4.1 INSTRUCTION TUNING

**Implementation Details.** We fine-tune the Mistral-7B, Gemma-7B, and Llama-3 8B models using the MetaMathQA (Yu et al., 2023a) dataset. This extensive dataset is derived from various complex mathematical instruction datasets, such as GSM8K and MATH, encompassing a wide range of diverse and challenging problem types. During the fine-tuning process, we utilize a subset of 100,000 questions from this dataset. To comprehensively evaluate the performance advantages of our LoRA adapter, we compare it with methods possessing a similar number of parameters, including LoRA (Hu et al., 2021b), DoRA (Liu et al., 2024a), VeRA (Liu et al., 2023a), and LoRA-XS (Zhang et al., 2023a). Subsequently, we assess these models on the validation sets of the GSM8K and MATH datasets, which feature intricate mathematical reasoning problems, thus providing an ideal context for evaluating the models' abilities in instruction adherence and logical reasoning.

**Comparison Results.** Table 1 presents the performance evaluation of the Mistral-7B, Gemma-7B, and Llama-3 8B models utilizing the NoRA method, demonstrating significant performance improvements. It is noteworthy that NoRA achieves an average performance improvement of over 4.4%, 2.5%, and 3.3% on the GSM8K and MATH datasets, respectively, compared to LoRA with a modest training parameter configuration of 0.92M across the three models.

## 4.2 FINE-TUNING OF LARGE LANGUAGE MODELS

**Implementation Details.** We employ a series of parameter-efficient methods to fine-tune the LLaMA-3 8B model (Yeh et al., 2023; Zhang et al., 2023b; Hayou et al., 2024; Valipour et al., 2022; Zhang et al., 2023a; Liu et al., 2023a), with the aim of enhancing its commonsense reasoning capabilities. Targeted fine-tuning is conducted using the Commonsense170K dataset to improve the model's comprehension of commonsense knowledge across diverse contexts. Subsequently, we evaluate the effectiveness of each fine-tuning method by assessing its impact on performance across various commonsense reasoning tasks. As a comparative approach to NoRA, techniques such as AdaLoRA (Zhang et al., 2023b) and DoRA Liu et al. (2024b) are applied to fine-tune the baseline model, which is then assessed using eight benchmarks emphasizing commonsense reasoning, including ARC-e, OBQA, SIQA, and others.

**Comparison Results.** Experimental evaluations, detailed in Table 2, reveal varying degrees of success among different fine-tuning methods aimed at enhancing the reasoning capabilities of the LLaMA-3 8B model. Notably, the NoRA approach emerges as a standout performer, achieving the highest average accuracy of 84.4%. It excels in specific tasks, securing top scores in HellaSwag (93.9%), WinoGrande (85.2%), and ARC-e (90.0%), demonstrating robust understanding and reasoning abilities across diverse question sets. NoRA's efficiency is further underscored by its utilization of significantly fewer parameters (4.1M) compared to resource-intensive methods like LoRA and AdaLoRA (28.3M), all without compromising competitive performance. These results highlight NoRA's high accuracy and enhanced parameter efficiency, making it an appealing choice for fine-tuning large pre-trained models, particularly in scenarios with limited computational resources.

Table 2: Average accuracy (%) on LLaMA-3 8B for 8 zero-shot tasks. #Params denotes the number of trainable parameters.

| Method | #Params | BoolQ | PIQA | SIQA | HellaSwag | WinoGrande | ARC-e | ARC-c | OBQA | Avg. |
|---|---|---|---|---|---|---|---|---|---|---|
| LoRA (2021b) | 28.3M | 72.3 | 86.7 | 79.3 | 93.5 | 84.8 | 87.7 | 75.7 | 82.8 | 82.8 |
| LoKr (2023) | 0.9M | 65.1 | 81.6 | 78.7 | 92.0 | 82.1 | 89.2 | 76.7 | 80.9 | 80.9 |
| AdaLoRA (2023b) | 28.3M | 75.1 | 86.4 | 76.7 | 75.4 | 83.3 | 90.4 | 79.1 | 81.4 | 81.4 |
| LoRA+ (2024) | 28.3M | 73.3 | 86.4 | 79.1 | 94.1 | 84.3 | 88.2 | 77.5 | 81.8 | 83.1 |
| DyLoRA (2022) | 29.1M | 71.4 | 86.1 | 79.4 | 91.7 | 81.9 | 90.1 | 78.8 | 82.4 | 82.8 |
| LoRA-FA (2023a) | 15.6M | 73.1 | 87.0 | 79.6 | 93.2 | 84.3 | 86.2 | 74.6 | 83.0 | 82.7 |
| VeRA (2023a) | 1.49M | 64.3 | 86.3 | 74.0 | 87.0 | 69.0 | **92.8** | **82.3** | 82.0 | 79.7 |
| DoRA (2024b) | 16.3M | 72.1 | **88.4** | **80.3** | 88.7 | **85.8** | 90.3 | 78.9 | **86.0** | 83.8 |
| NoRA | 4.1M | **74.0** | 87.4 | 80.0 | **93.9** | 85.2 | 90.0 | 79.7 | 84.6 | **84.4** |

Table 3: Detailed results for 5 datasets with the ViT-B/16 as visual backbone. Top-1 accuracy averaged over 3 random seeds is reported. Highest value is highlighted in bold, and the second highest is underlined.

| | | | Shots 4 | | | | | | | Shots 16 | | | |
|---|---|---|---|---|---|---|---|---|---|---|---|---|---|
| Method | Food | Pets | DTD | UCF | Cars | Average | Method | Food | Pets | DTD | UCF | Cars | Average |
| CoOp (2022b) (4) | 83.5 | 92.3 | 58.5 | 78.1 | 73.4 | 77.2 | CoOp (2022b) (4) | 85.1 | 92.4 | 81.2 | 81.9 | 79.1 | 83.9 |
| CoOp (2022b) (16) | 84.5 | 92.5 | 59.5 | 77.6 | 74.4 | 77.7 | CoOp (2022b) (16) | 84.2 | 92.0 | 69.7 | 83.1 | 82.9 | 82.4 |
| CoCoOp (2022a) | 86.3 | 92.7 | 55.7 | 75.3 | 69.5 | 75.9 | CoCoOp (2022a) | 87.4 | 93.4 | 63.7 | 77.2 | 72.3 | 78.8 |
| TIP-Adapter-F (2022) | 86.5 | 91.9 | 59.8 | 78.1 | 74.1 | 78.1 | TIP-Adapter-F (2022) | 86.8 | 92.6 | 70.8 | 83.9 | 82.3 | 83.3 |
| CLIP-Adapter (2024) | 86.5 | 90.8 | 46.1 | 70.6 | 67.5 | 72.3 | CLIP-Adapter (2022) | 87.1 | 92.3 | 59.4 | 80.2 | 74.0 | 78.6 |
| PLOT++ (2022) | 86.5 | 92.6 | 62.4 | 79.8 | 67.5 | 77.8 | PLOT++ (2022) | 87.1 | 93.6 | 71.4 | 85.3 | 84.6 | 84.4 |
| KgCoOp (2023) | 86.9 | 92.6 | 58.7 | 77.6 | 69.5 | 77.1 | KgCoOp (2023) | 87.2 | 93.2 | 68.7 | 81.7 | 74.8 | 81.1 |
| TaskRes (2023b) | 86.0 | 91.9 | 60.1 | 76.2 | 76.0 | 78.1 | TaskRes (2023b) | 86.9 | 92.4 | 71.5 | 84.0 | 83.5 | 83.7 |
| MaPLe (2023) | 86.7 | **93.3** | 59.0 | 77.1 | 70.1 | 77.2 | MaPLe (2023) | 87.4 | 93.2 | 68.4 | 81.4 | 74.3 | 80.9 |
| ProGrad (2023) | 85.4 | 92.1 | 59.7 | 77.9 | 75.0 | 78.0 | ProGrad (2023) | 85.8 | 92.8 | 68.8 | 82.7 | 82.9 | 82.6 |
| CLIP-LoRA (2024) | 82.7 | 91.0 | 63.8 | 81.1 | 77.4 | 79.2 | CLIP-LoRA (2024) | 84.2 | 92.0 | 72.0 | 86.7 | 86.3 | 84.3 |
| LoRA+ (2024) | 84.4 | 92.8 | 64.1 | 75.6 | 71.3 | 77.6 | LoRA+ (2024) | 85.1 | 93.6 | 72.1 | 84.9 | 86.1 | 84.4 |
| AdaLoRA (2023b) | 85.6 | 92.8 | **66.2** | 81.6 | 76.4 | _80.5_ | AdaLoRA (2023b) | 85.9 | 93.7 | _72.8_ | 86.2 | _86.4_ | 85.0 |
| DyLoRA (2022) | _87.0_ | 92.4 | 64.9 | 80.8 | **77.5** | 80.5 | DyLoRA (2022) | _87.6_ | 93.0 | 72.7 | 86.7 | 84.5 | 84.9 |
| LoRA-FA (2023a) | 86.7 | 93.0 | 64.4 | 80.1 | 77.2 | 80.3 | LoRA-FA (2023a) | 87.4 | _93.9_ | 71.9 | _86.9_ | 86.0 | _85.2_ |
| VeRA (2023a) | 84.5 | 92.5 | 65.1 | 81.3 | 77.1 | 80.1 | VeRA (2023a) | 86.2 | 92.2 | 72.2 | 86.1 | 85.3 | 84.4 |
| NoRA | **87.1** | _93.1_ | _65.2_ | 81.6 | _77.4_ | **80.9** | NoRA | **87.8** | **94.1** | **74.3** | **87.4** | **86.7** | **86.1** |

## 4.3 FINE-TUNING OF VISION-LANGUAGE MODELS

**Implementation Details.** Following the approach of previous work (Zanella & Ben Ayed, 2024), we evaluated various adaptation techniques on the Vision Transformer model (ViT-B/16) across five distinct datasets: Food101 (Bossard et al., 2014), OxfordPets (Parkhi et al., 2012), DTD (Cimpoi et al., 2014), UCF101 (Soomro et al., 2012), and StanfordCars (Krause et al., 2013). These datasets were selected to assess the robustness and adaptability of the methods across different visual domains. To ensure the reliability of the results, Top-1 accuracy was used as the primary performance metric, calculated as the average over three random seeds. Additionally, experiments were conducted under 4-shot and 16-shot settings to evaluate the effectiveness of each adaptation technique under conditions of limited data.

**Comparison Results.** Table 3 presents the Top-1 accuracy for each method across the five datasets under 4-shot and 16-shot settings. Notably, the NoRA model consistently outperforms other adaptation methods, demonstrating superior adaptability and efficiency. In the 4-shot setting, NoRA achieves an average Top-1 accuracy of 81.8, slightly exceeding DyLoRA, the second-best method. In the 16-shot setting, NoRA further excels, achieving an average Top-1 accuracy of 85.4, surpassing DyLoRA's score of 85.0. NoRA demonstrates exceptional robustness across visual domains, securing the best results in all individual datasets.

## 4.4 SUBJECT-DRIVEN GENERATION

**Implementation Details.** We investigate theme-based image generation utilizing advanced text-to-image diffusion models. A pre-trained text-to-image model is fine-tuned with images and specific textual prompts (e.g., "[V] photo of a cat") employing LoRA and NoRA adaptation techniques. The SDXL5 model (Podell et al., 2023) is fine-tuned on a 32G V100S GPU with a learning rate of $1 \times 10^{-4}$, a batch size of 4, and 500 training steps, which takes approximately 24 minutes.

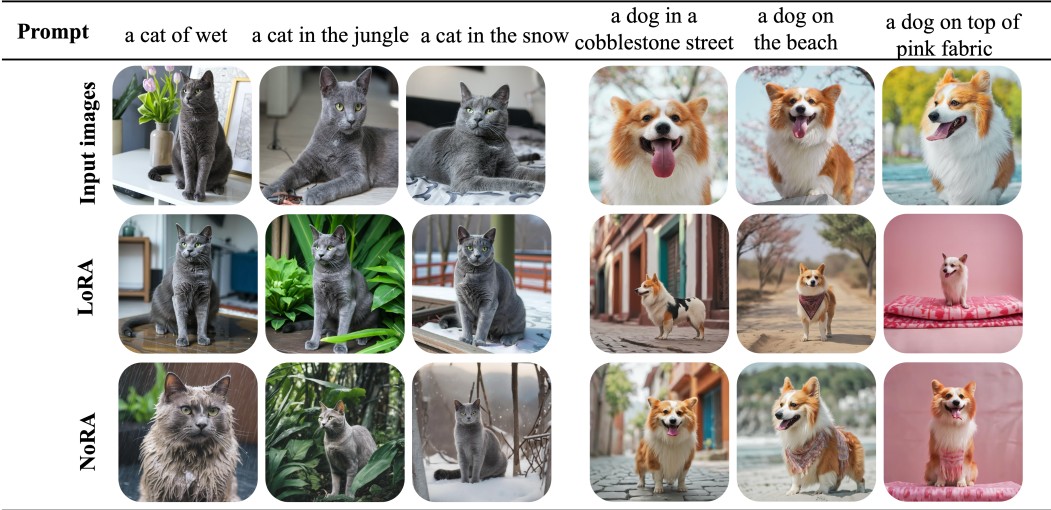

Figure 4: Comparative visualization of LoRA and NoRA performance on subject-driven image generation task. The illustration demonstrates the benefit of NoRA for models that adapt input images based on diverse prompts (e.g., "cat in the jungle" or "dog on the beach"), emphasizing the maintenance of thematic consistency and the accurate representation of diverse environments.

**Comparison Results.** Figure 4 presents the outcomes of the image generation task, utilizing 50 inference steps for each textual prompt. Compared to LoRA, the NoRA method demonstrates superior performance in capturing complex themes and intricate details, exhibiting enhanced visual alignment with the specified prompts. This improvement indicates greater thematic consistency and visual expressiveness. The advancements in image generation reveal significant potential for applications requiring detailed, context-specific imagery, thereby establishing a robust foundation for further exploration of fine-tuning techniques for complex thematic prompts.

Table 4: Ablation results on different initialization methods for **outer NoRA matrices** $W_B$ **and** $W_A$, applied to Mistra-8B across three experiments with different seeds.

| Initialization Methods | GSM-8K | MATH | AVG |
|---|---|---|---|
| Random | 66.4 | 17.1 | 41.8 |
| SVD | 69.1 | 18.9 | 44.0 |
| AwSVD | 69.4 | 19.1 | 44.3 |

Table 5: Ablation results on different initialization methods for **inner NoRA matrices** $w_b$ **and** $w_a$. The terms "Unif." and "Normal." represent the methods via uniform distribution and Gaussian distribution, respectively.

| Inner LoRA Init. | GSM-8K | MATH | AVG |
|---|---|---|---|
| Unif. ‖ Zero | 68.3 | 18.1 | 43.2 |
| diag($\Sigma_r$) ‖ diag($\Sigma_r$) | 68.9 | 18.8 | 43.9 |
| Normal. ‖ Normal. | 69.4 | 19.1 | 44.3 |

Table 6: Ablation results on different types for **inner NoRA matrix** $w_a$, applied to Mistra-8B across three experiments with different seeds.

| Type | GSM-8K | MATH | AVG |
|---|---|---|---|
| Adapter | 68.0 | 17.9 | 43.0 |
| LoRA Parallel | 66.4 | 17.4 | 41.9 |
| LoRA Serial | 69.4 | 19.1 | 44.3 |

Table 7: Ablation results on different LoRA serial position for **inner NoRA matrices** $w_b$ **and** $w_a$, applied to Mistra-8B across three experiments with different seeds.

| Location | GSM-8K | MATH | AVG |
|---|---|---|---|
| $w_a$ & $w_b$ | 68.9 | 18.7 | 43.8 |
| $w_b$ | 68.4 | 18.2 | 43.3 |
| $w_a$ | 69.4 | 19.1 | 44.3 |

## 4.5 ABLATION STUDY

**Initialization Strategies.** We compare various initialization methods, including random, SVD, and AwSVD, for the outer LoRA matrices $W_A$ and $W_B$ in Table 4. The ablation results indicate that AwSVD achieves the highest average performance on the GSM-8K and MATH datasets, with scores of 69.4 and 19.1, respectively. AwSVD effectively reduces SVD approximation errors while preserving the knowledge of the pre-trained model. For the initialization of inner NoRA matrices, we evaluate the performance of three methods: Gaussian distribution, diagonal singular matrix, and uniform initialization. As shown in Table 5, the Gaussian distribution yields superior performance, surpassing the other two methods.

**Structure Design Analysis.** Table 6 demonstrates that the serial LoRA method exhibits higher task accuracy compared to both parallel LoRA and adapter methods. Furthermore, Table 7 shows that applying the serial LoRA method exclusively at the $w_a$ position results in improved performance. Based on these findings, we derive NoRA design guidelines that emphasize the use of serial structures, design layouts, and nested LoRA.

**Training Time and Memory Usage.** In evaluating LoRA, DoRA, and NoRA on a commonsense reasoning task with controlled rank, NoRA displays superior efficiency in training time across different batch sizes. As shown in Figure 5 (a) and (b),, at a batch size of 4, NoRA is approximately 11 hours faster per batch than DoRA and 12 hours faster than LoRA. Additionally, NoRA demonstrates reduced GPU memory usage, particularly at larger batch sizes, indicating enhanced memory management and efficiency.

**Training Convergence Analysis.** Figure 5 illustrates NoRA's superior performance in terms of training loss compared to DoRA. NoRA rapidly converges to a lower loss value, with the curve steeply declining within the first 200 steps and maintaining a lower plateau throughout training, suggesting faster convergence and potentially more stable and effective training outcomes.

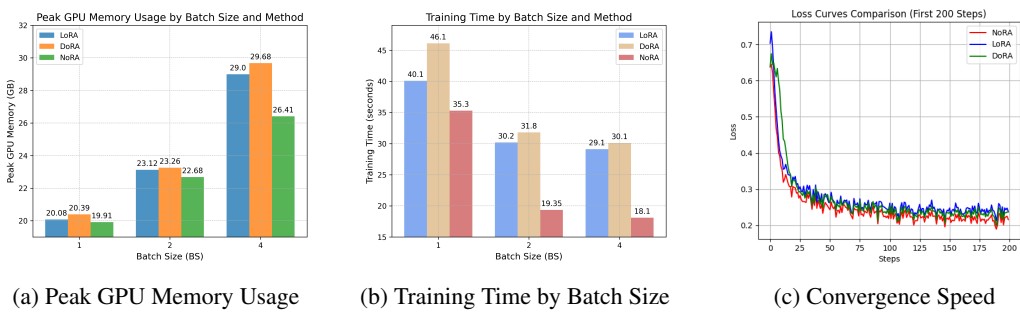

(a) Peak GPU Memory Usage   (b) Training Time by Batch Size   (c) Convergence Speed

Figure 5: Comparative Analysis of LoRA, DoRA, and NoRA

## 5 CONCLUSION

In this study, we introduce NoRA, an innovative framework for parameter-efficient fine-tuning that enhances the efficiency and effectiveness of LoRA-based methods. By establishing a unified design space, our comprehensive empirical analysis yields critical insights into initialization strategies, structural configurations, and design placements. Furthermore, we present the activation-aware SVD, which significantly reduces output errors and accelerates the training process. Comparative experiments across 15 datasets and 5 models demonstrate that NoRA not only preserves the parameter efficiency advantages of LoRA but also markedly improves overall performance. Future research may explore the integration of NoRA with AutoML and distillation techniques, applying it to multimodal models, and examining its effects on model interpretability and robustness.

**Limitations.** While NoRA shows strong performance across various tasks, its optimal hyperparameter configurations may vary depending on the specific task and models. This limitation is common and widespread in other LoRA variants and parameter-efficient fine-tuning methods.

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

APPENDIX

Our appendix provides supplementary information to the main paper, offering in-depth insights into our experimental procedures, extended discussions, and detailed setup configurations. It is organized into three main sections: (1) Extended Discussion, which elaborates on the differences between NoRA and existing work, acknowledges limitations, and considers potential societal impacts; (2) More Detailed Experiments, which presents additional results from our motivation experiments and extended NLP tasks; and (3) Experimental Setup and Hyperparameters, which outlines the specific configurations, hardware, software, and hyperparameters used in our studies. This comprehensive appendix aims to provide researchers with the necessary information to understand and potentially reproduce our results.

# A  MORE DISCUSSIONS

## A.1  ETHICS STATEMENT

This research focuses exclusively on developing efficient techniques for Large Language Models (LLMs), utilizing publicly available datasets and models. The study does not directly address human ethics or privacy concerns. Instead, it aims to enhance the computational efficiency and adaptability of existing LLMs, which may indirectly contribute to their broader accessibility and application.

## A.2  REPRODUCIBILITY

The authors affirm the solid reproducibility of their results and provide specific code implementations in the appendix. The main experiments represent average outcomes from multiple repetitions, ensuring reliability and consistency. By presenting detailed results for different initial seeds, the researchers demonstrate the robustness and repeatability of their method across various conditions, further solidifying the reproducibility of their findings.

## A.3  SUMMARY OF INNOVATIONS

(1) The study introduces NoRA, a novel nested parameter-efficient Low-Rank Adaptation (LoRA) design structure that optimizes the initialization and fine-tuning strategies of projection matrices. (2) The researchers propose an activation-aware Singular Value Decomposition (AwSVD) technique that adjusts weight matrices based on activation distributions, effectively managing outliers and accelerating model convergence. (3) The work constructs a unified design space for LoRA variants and develops comprehensive design guidelines, emphasizing the importance of specific design positions, serial structures, and the use of nested LoRA for enhanced performance and efficiency.

## A.4  PERFORMANCE GAINS

As the first nested LoRA method utilizing activation-aware SVD, NoRA demonstrates significant advantages in both performance and efficiency. (1) The performance gains compared to other LoRA variants are substantial, with NoRA achieving an average score of 84.4% on the LLaMA-3 8B model, surpassing LoRA's 82.8%. (2) In visual few-shot tasks, NoRA achieves the highest average accuracies of 80.9% (4 shots) and 86.1% (16 shots), outperforming existing methods. (3) The improvements in inference speed and memory optimization are notable strengths of NoRA, reducing the required parameters to as low as 4.1 million for the LLaMA-3 8B model while enhancing performance.

## A.5  COMPARISON TO OTHER METHODS

(1) While other LoRA variants like AdaLoRA, LoRA-FA, VeRA, and LoRA-XS have made advancements in low-rank adaptation, NoRA distinguishes itself by addressing key limitations in existing approaches. The unified design space and nested structure of NoRA offer unique advantages in balancing parameter efficiency and task-specific adaptation. Unlike methods that focus solely on rank adjustment or activation memory reduction, NoRA's comprehensive approach to optimization, including its AwSVD technique and nested structure, provides a more holistic solution to the challenges of fine-tuning large language models.

## A.6 SOCIETAL IMPACTS

The development of NoRA has potential societal implications: (1) Democratization of AI: By reducing computational requirements, NoRA could make fine-tuning large models more accessible to researchers and organizations with limited resources. (2) Environmental Benefits: Increased efficiency in model adaptation could lead to reduced energy consumption and carbon footprint associated with AI research and deployment.

# B MORE DETAILED EXPERIMENTS

## B.1 MOTIVATION EXPERIMENT RESULTS

Our motivation experiments focused on comparing different initialization strategies and architectural configurations. Key findings include:

- Figure 6 illustrates a subset of the structures within our unified design framework.

- SVD vs. Random Initialization: As shown in Table 8, SVD consistently outperformed random initialization across all tested datasets. For instance, in the Fine-tuning Vision-Language Models task, the maximum difference in average accuracy between SVD initialization and random initialization across the five datasets is 0.69 and 0.58 for 4-shot and 16-shot scenarios, respectively.

- AwSVD Performance: As shown in Figure 7, the Activation-aware SVD (AwSVD) method further improved upon standard SVD, showing about 10% reduction in output errors.

- Architectural Configurations: As shown in Table 9, the CLIP model with LoRA serial configuration outperforms the parallel configuration on diverse datasets. The average performance improvement is 2.5% and 2.55% for 4-shot and 16-shot, respectively. Additionally, compared to the adapter architecture, the LoRA serial configuration reduces the number of trainable parameters by 94%, leading to a more efficient parameter utilization.

Table 8: Detailed results for 5 datasets with the ViT-B/16 as visual backbone. Top-1 accuracy averaged over 3 random seeds is reported. Highest value is highlighted in bold, and the second highest is underlined.

| | Shots 4 | | | | | | | Shots 16 | | | | | |
|---|---|---|---|---|---|---|---|---|---|---|---|---|---|
| (WA,WB) | Food | Pets | DTD | UCF | Cars | Average | (WA,WB) | Food | Pets | DTD | UCF | Cars | Average |
| Random, Random | 85.94 | 93.24 | 64.07 | 79.25 | 73.61 | 79.22 | Random, Random | 87.12 | 94.33 | 71.28 | 86.02 | 84.72 | 84.69 |
| U$\Sigma$, V | 87.02 | 93.70 | 63.77 | 79.12 | 73.39 | 79.40 | U$\Sigma$, V | 87.60 | 94.49 | 72.70 | 86.12 | 85.46 | 85.27 |
| U, $\Sigma$V | 86.69 | 93.59 | 64.89 | 79.75 | 74.65 | 79.91 | U, $\Sigma$V | 87.44 | 94.25 | 72.64 | 86.62 | 84.72 | 85.13 |
| U$\sqrt{\Sigma}$, $\sqrt{\Sigma}$V | 86.81 | 93.92 | 64.18 | 79.28 | 73.78 | 79.59 | U$\sqrt{\Sigma}$, $\sqrt{\Sigma}$V | 87.56 | 94.17 | 72.40 | 86.41 | 85.01 | 85.11 |

Table 9: Detailed results for 5 datasets with the ViT-B/16 as visual backbone. Top-1 accuracy averaged over 3 random seeds is reported. Highest value is highlighted in bold, and the second highest is underlined. #Param represents the number of trainable parameters.

| | | Shots 4 | | | | | | | | Shots 16 | | | | | |
|---|---|---|---|---|---|---|---|---|---|---|---|---|---|---|---|
| $w_a$ | #Param | Food | Pets | DTD | UCF | Cars | Average | $w_a$ | #Param | Food | Pets | DTD | UCF | Cars | Average |
| LoRA Serial | 0.59M | 87.02 | 93.65 | 66.61 | 79.73 | 74.10 | 80.22 | LoRA Serial | 0.59M | 87.74 | 94.33 | 72.40 | 86.70 | 87.25 | 85.68 |
| LoRA parallel | 0.38M | 85.44 | 93.38 | 62.35 | 74.86 | 72.57 | 77.72 | LoRA parallel | 0.38M | 86.30 | 94.36 | 70.57 | 85.09 | 79.31 | 83.13 |
| Adapter Serial | 10.62M | 86.21 | 88.36 | 63.53 | 77.35 | 73.64 | 77.82 | Adapter Serial | 10.62M | 86.80 | 94.06 | 70.80 | 85.70 | 83.24 | 84.27 |

## B.2 ADDITIONAL NLP EXPERIMENT RESULTS

Extended results for natural language processing tasks:

- Based on the data in the table, we compared the performance of LoRA and NoRA methods on commonsense reasoning tasks using the LlaMA 7B model. Notably, NoRA demonstrated strong performance across multiple tasks, achieving an average score of 75.8%, which is slightly higher than LoRA's scores of 74.4% (r=16) and 75.3% (r=32).

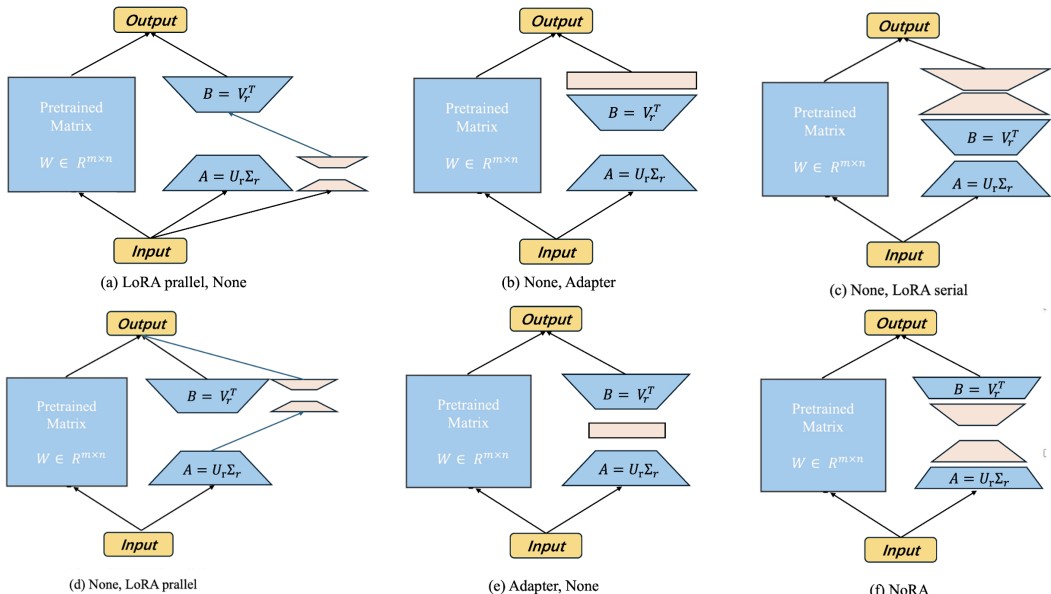

Figure 6: A subset of configurations within the unified design space $(w_a, w_b)$.

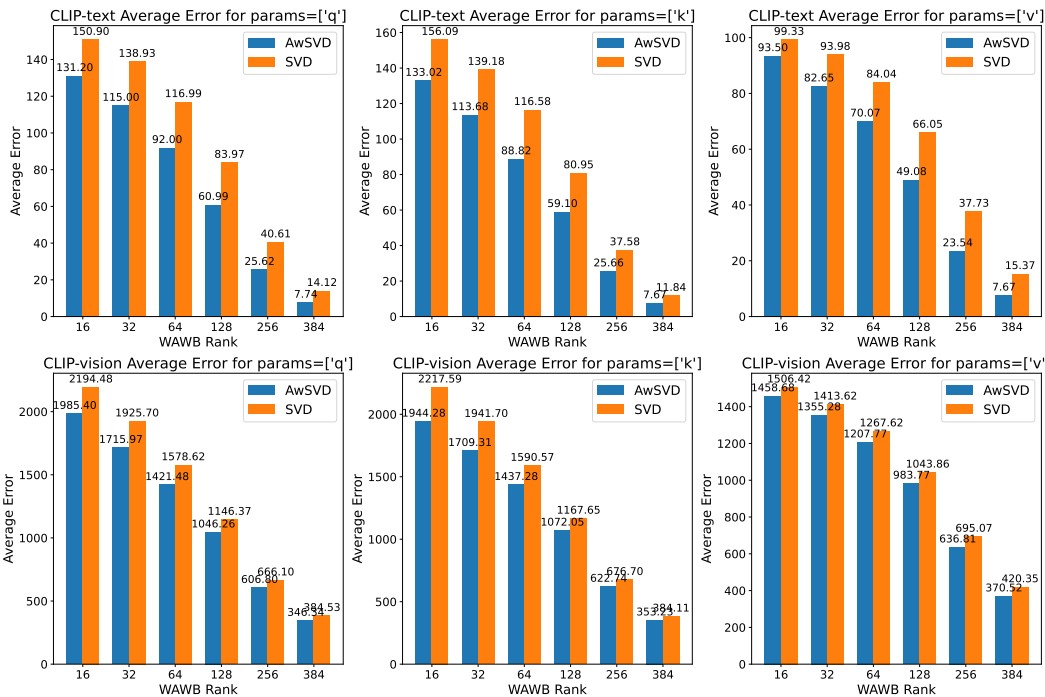

Figure 7: Comparison of SVD decomposition errors in CLIP text-encoder and vision-encoder across query projection, key projection, and value projection.

- Question Natural Language Inference: QNLI (Question Natural Language Inference) is a task from the GLUE (General Language Understanding Evaluation) benchmark. Using the QNLI dataset, NoRA achieved an accuracy of 94.6%, compared to 94.8% for LoRA and 94.7% for full fine-tuning, while reducing trainable parameters by 91% compared to LoRA and by 99.8% compared to full fine-tuning (see Table 10).

Table 10: GLUE Benchmark.

| Method | Trainable Parameters | QNLI |
|--------|----------------------|------|
| Full FT | 355M | 94.7 |
| LoRA | 800K | 94.8 |
| NoRA | 70K | 94.6 |

Table 11: Commonsense reasoning on LlaMA 7B

| Model | Method | BoolQ | PIQA | SIQA | HellaSwag | WinoGrande | ARC-e | ARC-c | OBQA | Avg |
|-------|--------|-------|------|------|-----------|------------|-------|-------|------|-----|
| | $LoRA_{r=16}$ | 68.9 | 80.7 | 77.4 | 78.1 | 78.8 | 77.8 | 61.3 | 74.8 | 74.4 |
| LlaMA 7B | $LoRA_{r=32}$ | 68.5 | 81.0 | 77.4 | 77.1 | 79.0 | 77.8 | 63.3 | 77.9 | 75.3 |
| | **NoRA** | 68.1 | 80.3 | 76.8 | 80.6 | 79.6 | 80.5 | 62.6 | 77.8 | 75.8 |

## C EXPERIMENTAL SETUP AND HYPERPARAMETERS

### C.1 MODEL CONFIGURATIONS

- CLIP ViT-B/16 vision encoder: 86.19 Million parameters, 12 layers, 768 hidden size
- CLIP ViT-B/16 text encoder: 63.43 Million parameters, 12 layers, 512 hidden size
- LLaMA-3 8B: 8 billion parameters, 32 layers, 4096 hidden size
- Mistral-7B: 7 billion parameters, 32 layers, 4096 hidden size
- Gemma-7B: 7 billion parameters, 28 layers, 3072 hidden size

### C.2 HARDWARE AND SOFTWARE

- GPUs: 8 x NVIDIA V100S (32GB)
- Framework: PyTorch 1.10.0
- CUDA Version: 11.3

### C.3 HYPERPARAMETERS

**Instruction Tuning:** We perform the instruction tuning experiments on Mistral-7B-v0.1 (Jiang et al., 2023) , Gemma-7B (Team et al., 2024) and LLaMA-3 8B models. We use a batch size of 128 and train for 2 epochs on 100k samples of the MetaMathQA dataset. Models are evaluated on the GSM8K and MATH datasets. The learning rate is set to 7E-3 with the AdamW optimizer (Loshchilov & Hutter, 2017). The warmup ratio is 0.02, and a cosine learning rate scheduler is used. The parameter $\alpha$ for NoRA modules is always equal to the rank. In NoRA (0.92M), the Outer and Inner LoRA ranks are 64 and 32, respectively. We used $8 \times$ V100S 32GB GPUs for the finetuning

**Fine-tuning of Vision-Language Models:** Table 12 details our hyperparameter settings for CLIP ViT-B/16, which remain consistent across all 5 datasets.

Common hyperparameters across experiments:

- Batch size: 32
- Learning rate: 1e-4 (AdamW optimizer)
- Weight decay: 0.01
- Warmup steps: 500
- Max steps: 20,000

Task-specific adjustments:

- GSM8K and Math: Increased max steps to 30,000
- Few-shot CLIP: Reduced batch size to 16, max steps to 5,000

Table 12: Our hyperparameter configuration on fine-tuning of Vision-Language model experiments.

| Hyperparameters | LoRA Serial |
|---|---|
| Batch size | 64 |
| Learning rate | 5e-4 |
| Scheduler | CosineAnnealingLR |
| Optimizer | AdamW |
| Weight decay | 0.01 |
| Dropout rate | 0.25 |
| Placement | query, key, value |
| n_iters | 400 |
| $(W_B, W_A)$ Init. | $(\mathbf{U\Sigma}, \mathbf{VS^{-1}})$ |
| Outer LoRA rank | 256 |
| Inner LoRA rank | 16 |

## C.4 EVALUATION METRICS

- NLP tasks: Accuracy, F1 score
- Math reasoning: Pass@1 score
- Few-shot image classification: Top-1 accuracy

