# OpenReview forum: "NoRA: Nested Low-Rank Adaptation for Efficient Fine-Tuning Large Models"
_ICLR.cc/2025/Conference — ICLR 2025 Conference Withdrawn Submission_

### Official Review · Reviewer_2psM · 2024-10-30

**Soundness:** 2
**Presentation:** 2
**Contribution:** 2
**Rating:** 3
**Confidence:** 4

**Summary:**

The paper introduces NoRA (Nested Low-Rank Adaptation), a parameter-efficient fine-tuning method for large models like Mistral-7B, Gemma-7B, and LLaMA-3 8B. It addresses the limitations of traditional LoRA (Low-Rank Adaptation), which involves tuning a large number of parameters, by proposing a nested structure that reduces parameter count while maintaining model adaptability and performance. Key contributions include:
- NoRA Architecture: A nested structure where outer LoRA layers are initialized using an activation-aware Singular Value Decomposition (AwSVD) to reduce decomposition errors, and inner LoRA layers are fine-tuned with fewer parameters, improving efficiency.
- AwSVD: An innovation that adjusts weight matrices based on activation distributions, ensuring higher fidelity to pre-trained weights and faster convergence during fine-tuning.
- Performance Improvements: NoRA significantly reduces fine-tuning parameters, memory usage, and training time while enhancing task performance. It outperforms other LoRA variants, achieving superior results across linguistic and visual tasks with fewer trainable parameters.

The paper demonstrates NoRA's efficiency through experiments, showing improvements in performance while reducing training-time and memory usage. The paper concludes by highlighting the advantages of NoRA in terms of expressiveness, flexibility, and parameter efficiency, positioning it as a robust method for fine-tuning large-scale models.

**Strengths:**

**1. Clear and Well-Structured Writing**

- The paper is well-written, with a clear and logical structure that makes it easy to follow. Concepts are explained in a straightforward manner, and the overall organization helps the reader grasp the technical content effectively.
- Figures and illustrations are clean, well-labeled, and support the text, helping to visually convey the architecture and results clearly.

**2. Innovative Techniques for Performance Improvement**

- The proposed nested LoRA structure, combined with the activation-aware Singular Value Decomposition (AwSVD) initialization, significantly enhances fine-tuning performance. This approach not only reduces the number of trainable parameters but also improves the model’s efficiency and adaptability to different tasks.

**3. Strong Performance Gains Over Comparable Methods**

- In comparison to other ultra-low parameter methods such as LoRA-XS and VeRA, NoRA demonstrates substantial performance improvements, particularly on challenging benchmarks like GSM8K and MATH. These results highlight the method's effectiveness in improving accuracy while maintaining parameter efficiency.

**Weaknesses:**

**1. Limited Scope of Comparative Analysis**

- The unified design space presented in the paper is not comprehensive enough. It primarily focuses on VeRA and LoRA-XS approaches, lacking coverage of other significant approaches in this domain.
- A comprehensive table summarizing the design choices of previous works is missing. Such a table would enhance the clarity and depth of comparisons.
- The comparison provided in Figure 2 resembles an ablation study of the proposed techniques rather than a thorough comparison of prior approaches. It would benefit from including more diverse methods in the analysis.

**2. Potential Compatibility with Other Approaches**

- There is no discussion of the compatibility of the proposed method with orthogonal approaches such as AdaLoRA and DoRA. Exploring how NoRA could integrate with or complement these methods could provide valuable insights.

**3. Theoretical and Intuitive Justifications**

- The paper introduces the concept of applying a scaling matrix to mitigate decomposition errors, but the intuition behind this approach is not clearly explained. A more rigorous theoretical justification is necessary.
- For AwSVD, it remains unclear whether this technique requires a large calibration set to perform effectively.

**4. Sensitivity and Practical Concerns**

- The results might be sensitive to batch size, but this aspect has not been explored in detail. A discussion on how batch size affects performance would strengthen the paper.
- It is also unclear how to select input activations from the fine-tuning dataset, which could impact the practical usability of the method.

**5. Rigor and Accuracy of Claims**

- Several statements lack rigor and precision. For instance, the claim regarding Formula 10 suggests that a tighter rank constraint leads to more complex non-linear transformations, but this assertion is misleading. A tighter rank constraint should not necessarily imply increased complexity.
- The explanation provided in line 308 on how to approximate the parallel LoRA form by rearranging terms is vague and needs further clarification.

**6. Issues with Reported Results**

- Table 1 only lists DoRA and LoRA as having rank=1 which is not a practical and feasible setting.
- Additionally, the accuracy of DoRA reported in Table 2 is inconsistent with the values in the original paper. The correct accuracy should be 85.3%, not 83.0%.
- A more holistic figure illustrating performance across different ranks would provide a clearer understanding of the flexible trade-off between expressiveness and efficiency.

**7. Novelty and Evaluation**

- The novelty of the proposed method could be questioned, as the core idea of SVD decomposition has already been explored and analyzed by LoRA-XS and PiSSA.
- The subject-driven image generation task appears selective, possibly bordering on cherry-picking. It would be beneficial to include qualitative results on widely used benchmarks, such as DreamBooth, to ensure a more objective evaluation.

**Questions:**

Already listed in the Weaknesses section

---

### Official Review · Reviewer_WBGK · 2024-10-31

**Soundness:** 1
**Presentation:** 1
**Contribution:** 1
**Rating:** 3
**Confidence:** 5

**Summary:**

This paper proposes a new PEFT method NoRA, as well as an initialization strategy.

**Strengths:**

The experiments are extensive.

The procedure of method is clear.

**Weaknesses:**

Too many weaknesses led me to choose to reject this paper. Furthermore, I believe this paper requires at least one major revision before it can be considered for a top-tier conference.

1. Line 43-44. “which can lead to slow convergence and potential overfitting problems.” The authors claimed two main approaches have emerged to address the aforementioned issues. However, DoRA cannot address them [1-2]. Indeed, DoRA sometimes easily fall into not converging.
2. Line 12, “but it still necessitates a substantial number of training parameters. To address this issue…”.  Line 53-86 “two significant challenges persist for these LoRA variants….To address these challenges…” In fact, I cannot understand why the authors conduct experiments in Fig.1 (a)(b). Even as the authors claimed in Line53-86, their experiment cannot answer problem 1, i.e., “the intrinsic properties of LLMs…..decomposition errors”
3. Line 248-254. It is better for the authors to provide more theoretical details on the design of activation-weight matrix and W_{aw}.
4. Line 281-290: Why NoRA’s rank allow for more complex non-linear transformations? I do not think this part “expressiveness” make sense. What can be concluded from the fact that the rank of NoRA is bounded by min(r, r′)? “Generation” part also makes no sense. And indeed, I believe the whole subsection 3.4 should be given further consideration.
5. I would like to see a comparison between LoRA and NoRA with similar parameter budgets.

Overall, the introduction of the paper is unclear, the motivation is not well-defined, the rationale behind the design in the methods section is unclear, and there are some issues with the effectiveness (Sec.3.4) of the approach.

[1] 2024, ICML, DoRA: Weight-Decomposed Low-Rank Adaptation

[2] 2024, arxiv, FLoRA: Low-Rank Core Space for N-dimension

**Questions:**

1. Line 186. ”PiSSA (Meng et al., 2024) selectively adjusts matrix ranks and distributions”. PISSA is a method focusing on initialization strategy. I would like authors to explain why PISSA can adjust matrix ranks, thanks.
2. Line 225. Why keeping the parameters of outer LoRA frozen can maintain stability? Could the authors provide theoretical justification or empirical evidence?
3. Line 224. “matrix B is initialized with UΣ”. What is U and Σ? It is the first time that these symbols appear, but they are not explained (as well as V and S).
4. Line 274-276. The adapter consists of two projection matrix and a non-linear(ReLU) layer. The adapter should be represented as Wx+B(ReLU(Ax)). Besides, what is a parallel LoRA? To transfer the matrix B to another LoRA layer? Please provide a clear definition and explanation.

---

### Official Review · Reviewer_MpXU · 2024-11-04

**Soundness:** 3
**Presentation:** 3
**Contribution:** 3
**Rating:** 5
**Confidence:** 4

**Summary:**

This paper presents a nested low-rank adaptation method for LLMs. NoRA employd an inner layer while freezing the outer layer to enable precise task-specific adaptations while maintaining compact training parameters. Extensive experiment results demonstrated its effectiveness.

**Strengths:**

+ The paper is well-structured and logically organized.
+ While some components are inspired by prior work, the integration of these elements is novel.
+ SoTA performance and low budgets.
+ Resonable motivations.

**Weaknesses:**

- The statement regarding NoRA’s rank enabling more complex non-linear transformations lacks theoretical grounding. The discussion around “expressiveness” in that section is underdeveloped. Simply stating that NoRA’s rank is limited by  \min(r, r{\prime})  does not elucidate how or why this rank impacts expressiveness. Thus, the authors should provide more experiments or theoretical explanations to demonstrate their claims.

- The decision to freeze the outer LoRA parameters to “maintain stability” lacks theoretical or empirical backing. Why this approach aids in stability needs elaboration？

- Comparative Analysis: A direct performance and parameter-efficiency comparison between LoRA and NoRA under similar parameter constraints would be insightful. This comparison would allow for a clearer understanding of how each approach performs relative to the other, given comparable resource budgets.

- In the visual section, the authors have conducted evaluations solely on a few simple classification datasets, which is insufficient. Additional complex vision-language tasks should be included, such as referring segmentation/detection, and visual caption tasks. This would better demonstrate the effectiveness of their proposed method.

**Questions:**

Please refer to weaknesses.

**Details Of Ethics Concerns:**

Please refer to weaknesses.

---

### Official Review · Reviewer_QfXB · 2024-11-04

**Soundness:** 4
**Presentation:** 4
**Contribution:** 3
**Rating:** 6
**Confidence:** 2

**Summary:**

Through a comprehensive empirical analysis, this paper provides critical insights into initialization strategies, structural configurations, and design placements. It further introduces an **Activation-Aware Singular Value Decomposition (AwSVD)** method to reduce output errors and accelerate the training process.

**Strengths:**

1. The paper is well-written and organized, with an intuitive motivation.
2. The method is clever, leveraging observations on LLMs—particularly their sensitivity to activation outliers—to propose an improved LoRA initialization. The upgrade from standard SVD to activation-aware SVD (AwSVD) enhances performance and reduces optimization difficulty.
3. The NORA structure, based on AwSVD initialization, further reduces the number of learnable parameters, enabling more efficient and lower-cost training. It’s also simple to implement, requiring only a few lines of code modifications, making it easily deployable and practical for application.

**Weaknesses:**

1. For instruction fine-tuning tasks, the paper only compares performance under settings with extremely low learnable parameters, which shows competitive results but falls significantly short of full-rank LoRA in performance. This raises concerns about whether the primary benefits of this work apply mainly to in-domain task transfers.
2. Is it necessary to reduce the number of optimization parameters in LoRA to save memory (particularly in the optimizer) or training time? After all, we don’t always need to compress parameter counts to such an extreme degree, and it often seems to be a trade-off. Unless it can be proven that this approach consistently outperforms various high- and low-rank LoRA fine-tuning methods, thereby serving as a superior replacement, the practical significance of this work remains in question.

**Questions:**

NA

---

### Official Review · Reviewer_Cb3x · 2024-11-06

**Soundness:** 2
**Presentation:** 2
**Contribution:** 2
**Rating:** 3
**Confidence:** 4

**Summary:**

The paper titled "NORA: NESTED LOW-RANK ADAPTATION FOR EFFICIENT FINE-TUNING LARGE MODELS" introduces a novel parameter-efficient fine-tuning method named NoRA (Nested Low-Rank Adaptation) for large language models (LLMs). NoRA addresses the challenge of high computational demands and training costs associated with traditional fine-tuning methods by optimizing the initialization and fine-tuning of projection matrices. The authors propose an activation-aware Singular Value Decomposition (AwSVD) technique to enhance the initialization process and reduce output errors. Extensive experiments across various linguistic and visual tasks demonstrate that NoRA outperforms existing Low-Rank Adaptation (LoRA) variants in terms of efficiency and effectiveness, significantly reducing fine-tuning parameters, training time, and memory usage while enhancing performance.

**Strengths:**

1、The paper provides a rigorous empirical evaluation of NoRA across a diverse set of linguistic and visual tasks, demonstrating its effectiveness and efficiency. The use of multiple benchmarks and the comparison against LoRA variants in various scenarios ensure that the results are robust and generalize well across different domains.
2、The paper employs a sound methodology, with a clear problem statement and a well-defined approach to address the challenges in fine-tuning large models. The activation-aware SVD (AwSVD) technique is a methodological innovation that leverages activation distributions for more accurate weight matrix initialization, which is a sophisticated approach to enhancing model performance.
3、 NoRA demonstrates a significant reduction in fine-tuning parameters, training time, and memory usage, which is a critical advantage in the context of large language models that typically require substantial computational resources. This resource efficiency makes NoRA particularly appealing for applications where computational budgets are limited.

**Weaknesses:**

1、The core of the article appears to revolve around the activation-aware matrix, which is the foundation and heart of the entire method. However, the paper seems to lack a discussion on how to confirm that the activation-aware matrix used is superior, whether there are other methods available, and how to determine whether this matrix can provide more useful information. Moreover, the approach of merely performing singular value decomposition on the activation-aware matrix and then nesting LoRA matrices might appear to offer limited innovation.
2、To better understand the contribution of each component of NoRA, such as the nested structure and AwSVD, the paper would benefit from ablation studies. These studies would isolate the effects of different design choices and provide insights into which aspects are most critical for the performance improvements observed.
3、The paper mentions that the optimal hyperparameter configurations for NoRA may vary depending on the specific task and models. This sensitivity could be a limitation for users who need to fine-tune models for different applications.

**Questions:**

The paper introduces NoRA as a two-layer nested low-rank adaptation structure. Have the authors considered exploring nested structures with more than two layers, and if so, what are the potential benefits or drawbacks? Could a deeper nested structure lead to improved performance, and if it does, is there a point of diminishing returns? Additionally, how does the computational complexity scale with the increase in the number of layers in the nested structure?

---

### Note · Authors · 2024-11-13

I have read and agree with the venue's withdrawal policy on behalf of myself and my co-authors.